# Structure of human phagocyte NADPH oxidase in the resting state

Rui Liu[1,2†], Kangcheng Song[1,2†], Jing-Xiang Wu[1,2], Xiao-Peng Geng[1,2], Liming Zheng[3], Xiaoyin Gao[3], Hailin Peng[3], Lei Chen[1,2,4]*

[1]State Key Laboratory of Membrane Biology, College of Future Technology, Institute of Molecular Medicine, Peking University, Beijing Key Laboratory of Cardiometabolic Molecular Medicine, Beijing, China; [2]National Biomedical Imaging Center, Peking University, Beijing, China; [3]Center for Nanochemistry, Beijing Science and Engineering Center for Nanocarbons, Beijing National Laboratory for Molecular Sciences, College of Chemistry and Molecular Engineering, Peking University, Beijing, China; [4]Peking-Tsinghua Center for Life Sciences, Peking University, Beijing, China

**Abstract** Phagocyte oxidase plays an essential role in the first line of host defense against pathogens. It oxidizes intracellular NADPH to reduce extracellular oxygen to produce superoxide anions that participate in pathogen killing. The resting phagocyte oxidase is a heterodimeric complex formed by two transmembrane proteins NOX2 and p22. Despite the physiological importance of this complex, its structure remains elusive. Here, we reported the cryo-EM structure of the functional human NOX2-p22 complex in nanodisc in the resting state. NOX2 shows a canonical 6-TM architecture of NOX and p22 has four transmembrane helices. M3, M4, and M5 of NOX2, and M1 and M4 helices of p22 are involved in the heterodimer formation. Dehydrogenase (DH) domain of NOX2 in the resting state is not optimally docked onto the transmembrane domain, leading to inefficient electron transfer and NADPH binding. Structural analysis suggests that the cytosolic factors might activate the NOX2-p22 complex by stabilizing the DH in a productive docked conformation.

*For correspondence:
chenlei2016@pku.edu.cn

†These authors contributed equally to this work

Competing interest: The authors declare that no competing interests exist.

## Editor's evaluation

NADPH oxidases are a family of membrane enzymes that produce reactive oxygen species (ROS). NOX2 is the most well-studied member of the NADPH oxidase family, and the proper function of NOX2 is critical for innate immunity against pathogens in mammals. The study by Dr. Chen and colleagues used antibodies to facilitate the structural determination of the high-resolution structure of the NOX2-p22 complex, which is otherwise challenging for single-particle analysis due to its flexibility and relatively small molecular weight. This structural study provides valuable information for a mechanistic understanding of NOX2 activation at the molecular level.

## Introduction

NADPH oxidases (NOX) are membrane-bound redox enzymes (*Lambeth and Neish, 2014*). They transfer electrons from intracellular NADPH to extracellular oxygen to generate reactive oxygen species, including superoxide anions and hydrogen peroxide (*Lambeth and Neish, 2014*). There are seven NOX members identified in humans, including NOX1-5 and DUOX1-2 (*Bedard and Krause, 2007*). They are involved in broad physiological processes, such as host defense, cell signaling, protein modification, and hormone synthesis (*Lambeth and Neish, 2014*). The mutations of genes encoding NOX proteins can cause human diseases such as immunodeficiency and hypothyroidism (*Bedard and Krause, 2007*).

NOX2 is the key catalytic subunit of phagocyte oxidase, which is highly expressed in phagocytes and is essential for innate immunity (*Winterbourn et al., 2016*). Resting phagocyte oxidase is a heterodimer composed of two multi-pass transmembrane proteins NOX2 (gp91phox) and p22phox (p22) (*Winterbourn et al., 2016*). Once invading pathogens are trapped inside the special membrane compartment named phagosomes, a phosphorylation-dependent signaling cascade activates the assembly of cytosolic factors onto NOX2, including p47phox, p67phox, p40 phox, and Rac (*Winterbourn et al., 2016*). Upon the binding of cytosolic factors onto the NOX2-p22 complex, NOX2 transfers electrons from intracellular NADPH to oxygen, causing the respiratory burst of phagocytes. The generated superoxide anions play crucial roles in killing the invading pathogens inside phagosomes (*Winterbourn et al., 2016*). In accordance with the key physiological function of NOX2, the loss-of-function mutations of the *NOX2* gene lead to failures of clearance of invading pathogens due to low levels of superoxide anions produced, causing chronic granulomatous disease (CGD) (*Heyworth et al., 2003*).

Previous studies show that the catalytic core of NOX has a transmembrane domain (TMD) and a cytosolic dehydrogenase (DH) domain (*Magnani et al., 2017*; *Sun, 2020*; *Wu et al., 2021*). TMD has six transmembrane helices which coordinate two haems for electron transfer across the membrane (*Magnani et al., 2017*; *Sun, 2020*; *Wu et al., 2021*). DH domain shares sequence homology with the ferredoxin-NADP reductase (*Magnani et al., 2017*; *Sun, 2020*; *Wu et al., 2021*). DH domain is formed by the FAD-binding sub-domain (FBD) and NADPH binding sub-domain (*Magnani et al., 2017*; *Sun, 2020*; *Wu et al., 2021*). The available crystal structure of the TMD of NOX5 from *Cylindrospermum stagnale* (csNOX5) revealed the electron transfer pathway across the membrane and the oxygen substrate binding site, also named oxygen-reducing center (ORC) (*Magnani et al., 2017*). Recent cryo-EM structures of the DUOX1-DUOXA1 complex showed not only how the FAD cofactor and NADPH substrate are sandwiched at the interface between DH domain and TMD of DUOX but also the complete electron transfer pathway from the intracellular NADPH to the extracellular oxygen (*Sun, 2020*; *Wu et al., 2021*). Moreover, the structure of human DUOX1-DUOXA1 in the high-calcium state revealed the conformation of the activated NOX (*Wu et al., 2021*). These studies have provided instrumental clues about the structure of NOX2. However, different from NOX5 or DUOX, the function of NOX2 protein absolutely requires an essential auxiliary subunit p22 (*Bedard and Krause, 2007*). p22 harbors the docking site for the cytosolic component p47phox protein which is crucial for the assembly of NOX2 with cytosolic components. The loss-of-function mutations of p22 also lead to CGD (*Heyworth et al., 2003*), emphasizing its physiological importance. Notably, p22 can also form a complex with NOX1, NOX3, and NOX4 and is essential for the functions of these proteins. Prior to our current study, the structure of p22 is largely unknown due to the lack of homology models, and the topology of p22 is controversial because different studies suggest that p22 has two or three, or four transmembrane helices (*Heyworth et al., 2003*; *Stasia, 2016*; *Meijles et al., 2012*; *Dahan et al., 2002*). More importantly, how p22 interacts with NOX1-4 protein remains mysterious. Here, we present the structure of the functional human NOX2-p22 complex in the resting state, uncovering the architecture of this important enzyme.

## Results
### Structure determination

To measure the superoxide anions released from the NOX2 enzyme, we first converted the superoxide anion into hydrogen peroxide with SOD. Then, we used the Amplex Red assay (*Wu et al., 2021*; *Forteza et al., 2005*) to measure the concentration of hydrogen peroxide (*Figure 1A*). This assay provided highly sensitive measurement of NOX activity in real time, which is crucial to detect NOX activity in cell lysates from small-scale culture (*Wu et al., 2021*). To activate NOX2 in a cell-free system, we exploited an elegantly designed chimeric protein named Trimera, which contains essential fragments of p47phox, p67phox, and Rac1 for NOX2 activation (*Berdichevsky et al., 2007*). Using this assay system, we found that co-expression of human NOX2 and C-terminally GFP-tagged human p22 could yield cell membranes that showed Trimera-activated superoxide anion production which is sensitive to diphenyliodonium (DPI) inhibition (*Figure 1B*), suggesting a correctly assembled NOX2-p22 hetero-complex in our expression system. However, subsequent purification and cryo-EM analysis of this complex only resulted in a medium-resolution reconstruction that was not sufficient

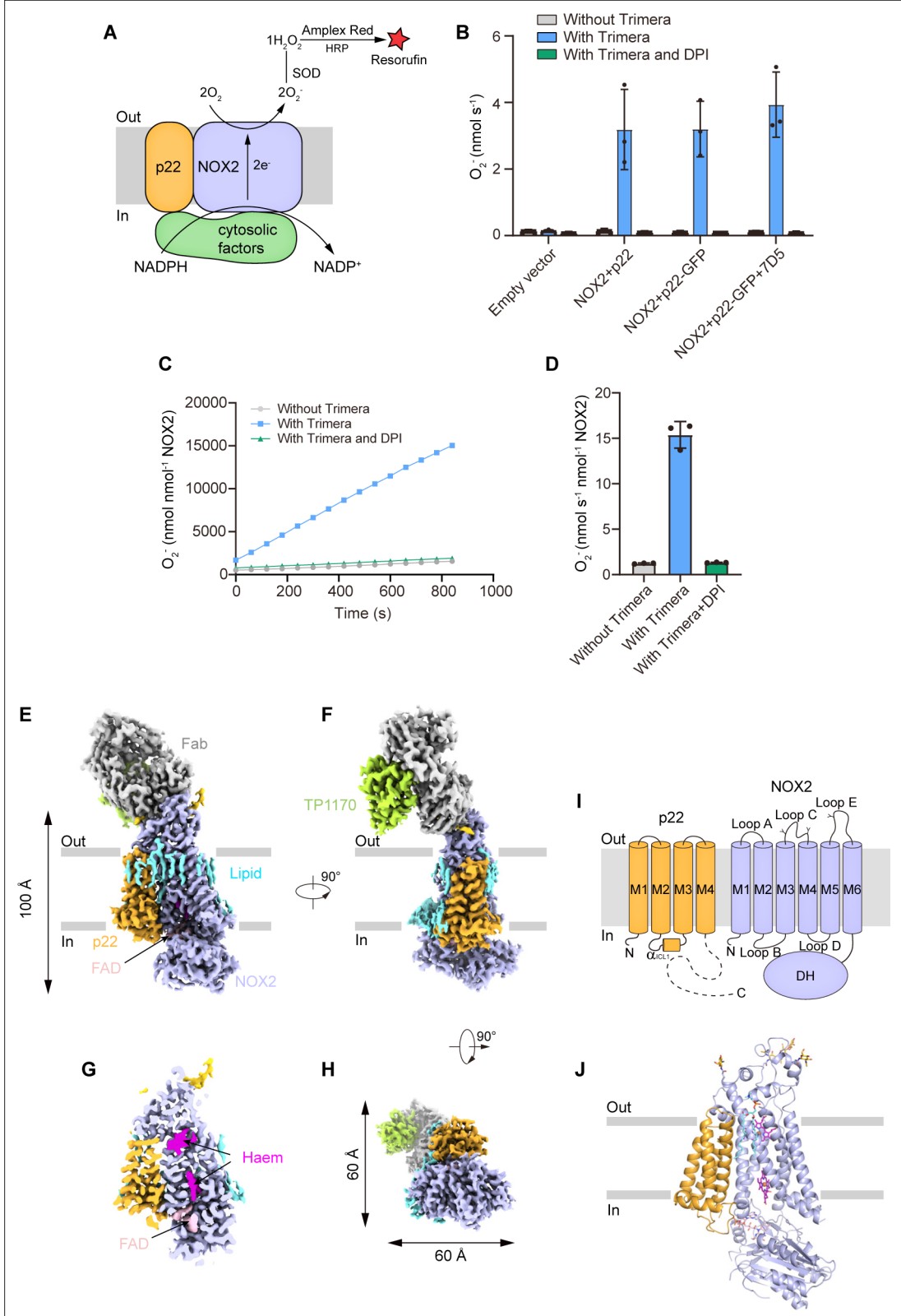

**Figure 1.** The structure of the human NOX2-p22 complex in the resting state. (**A**) Schematic of the NOX2 enzymatic assay. $O_2^-$ produced by NOX2 are converted into $H_2O_2$ by SOD. In the presence of $H_2O_2$, horseradish peroxidase (HRP) converts the nonfluorescent Amplex Red into resorufin, the fluorescence of which is measurable and proportional to the concentration of $H_2O_2$. (**B**) The activity of the NOX2-p22 complex in crude cell membrane measured using Amplex Red assay. The activity is determined by subtracting the background of the enzymatic assay buffer without crude cell

*Figure 1 continued on next page*

*Figure 1 continued*

membrane. Data are shown as means ± standard deviations; N=3 technical replicates. (**C**) The amounts of $O_2^-$ produced by NOX2-p22-7D5-TP1170 complex in nanodisc in the presence of Trimera are plotted versus time. The represented data are shown. DPI, diphenyliodonium, NADPH oxidase inhibitor. (**D**) The rates of $O_2^-$ production in C are summarized. Data are shown as means ± standard deviations; N=3 technical replicates. (**E**) Side view of the cryo-EM map of NOX2-p22-7D5-TP1170 complex. The approximate boundaries of the phospholipid bilayer are indicated as gray thick lines. p22 and NOX2 are colored in orange and light blue. 7D5 and TP1170 are colored in gray and light green. FAD, lipid, and glycosylation decoration are colored in pink, cyan, and gold, respectively. (**F**) A 90°-rotated side view compared to D. (**G**) The cut-open view of the cryo-EM map of NOX2-p22. Haem and FAD is colored in magenta and pink, respectively. (**H**) A 90°-rotated bottom view of E. (**I**) Topology of p22 and NOX2 subunits. Transmembrane helices are shown as cylinders, unmodeled disordered regions are shown as dashed lines. The phospholipid bilayer is shown as gray layers. Glycosylation sites were indicated by protrusions. DH, dehydrogenase domain of NOX2. (**J**) Structure of NOX2-p22 complex in cartoon representation. The colors are the same as in D.

The online version of this article includes the following source data and figure supplement(s) for figure 1:

**Figure supplement 1.** Protein purification.

**Figure supplement 1—source data 1.** The uncropped and unedited gels for *Figure 1—figure supplement 1B*.

**Figure supplement 2.** Image processing.

**Figure supplement 3.** Representative local electron density maps contoured at 4.62σ.

**Figure supplement 4.** Sequence alignment of the NADPH oxidases (NOX) subunit.

**Figure supplement 5.** Sequence alignment of p22 subunit.

**Figure supplement 6.** Structural alignment of NADPH oxidases (NOX) family proteins.

**Figure supplement 7.** Locations of chronic granulomatous disease (CGD) mutations found in human patients.

for confident model building. We reasoned that the small size of the complex and the endogenous conformational heterogeneity of NOX2 DH domain might represent major obstacles to our structural studies. Therefore, we exploited the possibility of using a commercially available mouse anti-human NOX2 monoclonal antibody 7D5 (*Yamauchi et al., 2001*; *Nakamura et al., 1987*; *Kawai et al., 2018*) as a fiducial marker for cryo-EM studies. We produced the Fab fragment of 7D5 by papain digestion. We found that 7D5 Fab does not affect the activity of NOX2 in the cell-free activity assay system (*Figure 1B*), suggesting it does not interfere with the electron-transfer pathway of the NOX2-p22 complex. To further increase the molecular weight of the NOX2-p22-7D5 Fab complex, we used strep-tagged anti-mouse kappa chain nanobody TP1170 to bind 7D5 Fab (*Pleiner et al., 2018*). We reconstituted the human NOX2-p22 hetero-complex in nanodisc and purified the NOX2-p22-7D5-TP1170 complex (*Figure 1—figure supplement 1A–B*). The resulting quaternary complex showed moderately active Trimera-dependent DPI-sensitive superoxide production (*Figure 1C–D*), indicating the protein sample is functional and suitable for structural studies. Cryo-EM data collection of the protein sample absorbed on both graphene-oxide-coated grids (*Patel et al., 2021*) and graphene-coated grids (*Zhang et al., 2017*) yielded a consensus reconstruction of this asymmetric complex to an overall resolution of 2.81 Å (*Figure 1—figure supplement 2*). However, the local map quality of the intracellular DH domain of NOX2 and the constant regions of Fab heavy chain and light chain and TP1170 on the extracellular side was poor in the consensus map due to their marked mobility. Subsequent signal subtraction and local refinement focusing on these two regions using mask 1 and mask 2 further generated 3.1 Å reconstructions and improved their local map quality (*Figure 1—figure supplement 2*). The map of the core region is also improved using this local refine strategy with mask 3 (*Figure 1—figure supplement 2*). We aligned three focused refined maps to the consensus map and merged them to generate one composite map for model building and interpretation (*Figure 1—figure supplement 2* and *Table 1*).

The final structural model contains most residues of NOX2 and 3–136 of p22 (*Figure 1E–J*, *Figure 1—figure supplements 4–5*). The long cytosolic tail of p22 is invisible in the cryo-EM map due to its high flexibility without cytosolic factors. The structures of TMD and DH domains of NOX2 are overall similar to the structure of csNOX5 and DUOX (*Magnani et al., 2017*; *Sun, 2020*; *Wu et al., 2021*; *Figure 1—figure supplement 6*). p22 has a four-helix bundle architecture and binds to the NOX2 TMD from the side, forming a hetero-complex occupying 100 Å × 60 Å × 60 Å in the three-dimensional (3D) space (*Figure 1E–H*). We mapped the spatial positions of NOX2 and p22 mutations found in CGD patients onto this structure (*Figure 1—figure supplement 7*). Moreover, we found that 7D5 Fab recognizes a structural epitope on the extracellular side of NOX2 (*Figure 1E–H*). TP1170

**Table 1.** Cryo-EM data collection, refinement, and validation statistics.

| PDB ID<br>EMDB ID | NOX2-p22-7D5-TP11708GZ3EMD-34389EMD-34390[*]/EMD-34620[†]/EMD-34622[‡]/EMD-34621[§] |
| --- | --- |
| **Data collection and processing** | |
| Magnification | ×165,000 |
| Voltage (kV) | 300 |
| Electron exposure (e⁻/Å²) | 37.6 |
| Defocus range (μm) | −1.5 to −2.0 |
| Pixel size (Å) | 0.821 |
| Symmetry imposed | *C1* |
| Initial particle images (no.) | 541,609 |
| Final particle images (no.) | 84,035 |
| Map resolution (Å) | 2.81[*]/3.1[†]/3.1[‡]/2.75[§] |
| FSC threshold | 0.143 |
| Map resolution range (Å) | 250–2.8 |
| **Refinement** | |
| Initial model used (PDB code) | Alpha Fold 2 |
| Model resolution (Å) | 2.8 |
| FSC threshold | 0.143 |
| Model resolution range (Å) | 250–2.8 |
| Map sharpening *B* factor (Å²) | −89.6[*]/−128.9[†]/−123.3[‡]/100.1[§] |
| **Model composition** | |
| Non-hydrogen atoms | 8,542 |
| Protein residues | 1,214 |
| Ligands | 9 |
| **_B_ factors (Å²)** | |
| Protein | 55.46 |
| Ligand | 58.94 |
| **R.m.s. deviations** | |
| Bond lengths (Å) | 0.011 |
| Bond angles (°) | 1.531 |
| **Validation** | |
| MolProbity score | 1.60 |
| Clashscore | 5.77 |
| Poor rotamers(%) | 1.74 |
| **Ramachandran plot** | |
| Favored (%) | 97.49 |
| Allowed (%) | 2.42 |
| Disallowed (%) | 0.08 |

[*]The values of consensus refinement.

[†]The values of focused refinement of mask1 (CH1 + CL + TP1170).

[‡]The values of focused refinement of mask2 (DH domain).

[§]The values of focused refinement of mask3 (core).

binds to the constant domain of 7D5 Fab kappa light chain (*Figure 1F*), which is consistent with previous functional data (*Pleiner et al., 2018*). In addition to protein densities, we observed many lipid-like densities (*Figure 1E–F*). Based on the shape and its chemical environment, we tentatively assign one lipid on the NOX2-p22 interface as a phosphatidylcholine molecule (*Figure 1—figure supplement 3C*). The identity and functions of other lipids await further investigation. Because NOX2 requires cytosolic factors for activation (reviewed in *Groemping and Rittinger, 2005*) and there are no cytosolic factors supplemented in the cryo-EM sample, the current structure represents the human NOX2-p22 complex in the resting state before activation.

## Structure of human NOX2

NOX2 resembles the canonical structure of NOX, which consists of an N-terminal TMD and a C-terminal cytosolic DH domain (*Figure 2A*). The extracellular region is formed by Loop A, Loop C, and Loop E (*Figure 2B*, *Figure 1—figure supplement 4*). Loop E sits on the top and there is a glycosylation modification at N240. An intra-loop disulfide bond (C244-C257) stabilizes Loop E in a compact structure (*Figure 2B*). Notably, C244R (*Rae et al., 1998*), C244S (*Bolscher et al., 1991*), and C244Y (*Patiño et al., 1999*) mutations on NOX2 were previously identified in CGD patients (*Figure 1—figure supplement 7*). These mutations likely affect the conformational stability of Loop E and thus the maturation of NOX2. We observed two N-link glycosylation decorations at N132 and N149 on Loop C (*Figure 2B*). Both Loop C and Loop E form the structural epitope that is recognized by 7D5 Fab (*Figure 1E–F*, *Figure 2B*). Loop A buttresses below Loop C and Loop E to support their conformation (*Figure 2B*). These three extracellular loops form a dome that caps above the outer haem bound in the TMD of NOX2 (*Figure 2A–B*). In our high-resolution map, we observe an ordered non-protein density which is surrounded by the highly conserved R54, H119, and the outer haem (*Figure 2C*). The density has also been previously observed in the crystal structure of csNOX5 (*Magnani et al., 2017*) and cryo-EM structure of human DUOX1 (*Wu et al., 2021*) and has been considered as a water molecule bound right at the ORC (*Figure 2C*). Using Caver (*Chovancova et al., 2012*), we identified a hydrophilic tunnel that connects the extracellular environment to the ORC (*Figure 2D–F*). The radius of the tunnel is sufficiently large for the permeation of oxygen and superoxide anion (*Figure 2F*). This tunnel might represent the putative pathway for the entry of oxygen substrate and release of superoxide anions. The outer haem, inner haem, and F215 between them form the electron-transferring pathway across the membrane (*Figure 2G*), in agreement with the previous studies (*Magnani et al., 2017*; *Sun, 2020*; *Wu et al., 2021*). The outer haem is coordinated by H115 on M3 and H222 on M5. The inner haem is coordinated by H101 on M3 and H209 on M5 (*Figure 2G*). Below the TMD, we observed a strong density of FAD which binds in the FBD of DH domain (*Figure 2H* and *Figure 1—figure supplement 3E*). The adenosine ring of FAD packs below F202 of M5. The flavin ring of FAD binds in a pocket formed by P339, F340, T341, and R356 (*Figure 2H*). In contrast to FAD, we did not observe a strong density of NADPH (*Figure 1—figure supplement 3*), although we have supplemented 1 mM NADPH into the cryo-EM samples, indicating the affinity for NADPH is low in the resting state of NOX2. The structure of NOX2 allowed us to measure the edge-to-edge distances as 3.7 Å between the outer haem (atom CBC) and F215 (atom CZ); 3.8 Å between F215 (atom CD2) and the inner haem (atom CBC); 6.1 Å between the inner haem (O2Z) and FAD (atom N1A) (*Figure 2D*).

## Structure of human p22

The structure of p22 consists of four tightly packed transmembrane helices (*Figure 3*). On the intracellular loop 1 (ICL1) between M2 and M3, an amphipathic helix $\alpha_{ICL1}$ floats on the inner leaflet of the membrane (*Figure 3A*). A polar interaction network, formed by side chains of N11 on M1, E53 on M2, R90 and H94 on M3, and Y121 on M4 is caged inside the TMD of p22 (*Figure 3A*). These residues are highly conserved in p22 (*Figure 1—figure supplement 5*) and are CGD mutation hotspots (*Figure 1—figure supplement 7*). It is reported that R90Q (*de Boer et al., 1992*), R90W (*Rae et al., 2000*), E53V (*Hossle et al., 1994*), and H94R (*de Boer et al., 1992*) mutations of p22 have been identified in CGD patients (*Figure 1—figure supplement 7*), indicating the polar interaction network formed by these residues is important for the function of p22. Below the TMD, the 128–136 segment of p22 extends out of M4 and binds into a crevice shaped by residues on $\alpha_{ICL1}$ and M2-$\alpha_{ICL1}$ linker (*Figure 3B*). This binding mode might position the p22 tail in a suitable pose that is ready for p47[phox] recruitment and subsequent assembly of other cytosolic factors during NOX2 activation.

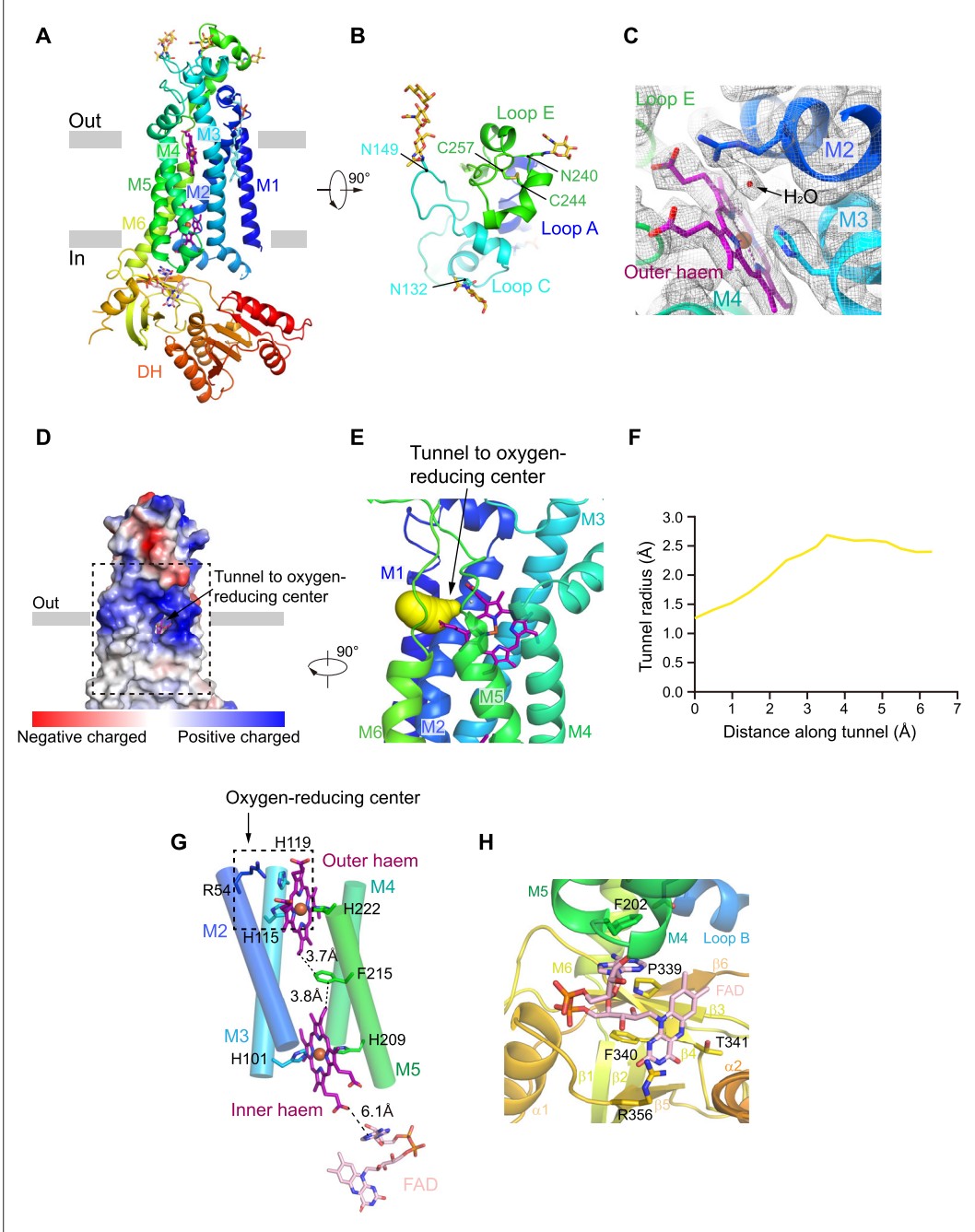

**Figure 2.** The structure of human NOX2. (**A**) Structure of NOX2 in cartoon representation, and is colored in the rainbow pattern (N-terminus is blue and C-terminus is red). The phospholipid bilayer is shown as gray layers. (**B**) A 90°-rotated top view of A. The disulfide bonds and glycosylation are shown as sticks. Three extracellular loops are indicated. (**C**) Electron density at the oxygen-reducing center contoured at 6.3σ. The density of water is indicated with an arrow. The color is the same as in A. (**D**) Surface electrostatic potential distribution of NOX2 estimated by pymol. The tunnel to the oxygen reducing center is indicated with an arrow. Haem and glycosylation are shown as sticks. The phospholipid bilayer is shown as gray layers. (**E**) Side view of the oxygen-reducing center as shown in a dashed box in D. The predicted tunnel for the entry of oxygen substrate and release of superoxide anions is colored in yellow and indicated with an arrow. (**F**) Calculated tunnel radius shown in E. The starting point for calculation is the oxygen-reducing center. (**G**) The components along the electron transfer pathway of NOX. The edge-to-edge distances between adjacent components are shown in dashes. The oxygen reducing center is indicated as a dashed rectangle. The ligands are shown as sticks. The ferric ions are shown as spheres. The helices are shown as cylinders. The colors are the same as in A. (**H**) The close-up view of the FAD-binding site. Side chains of residues interacting with FAD are shown as sticks.

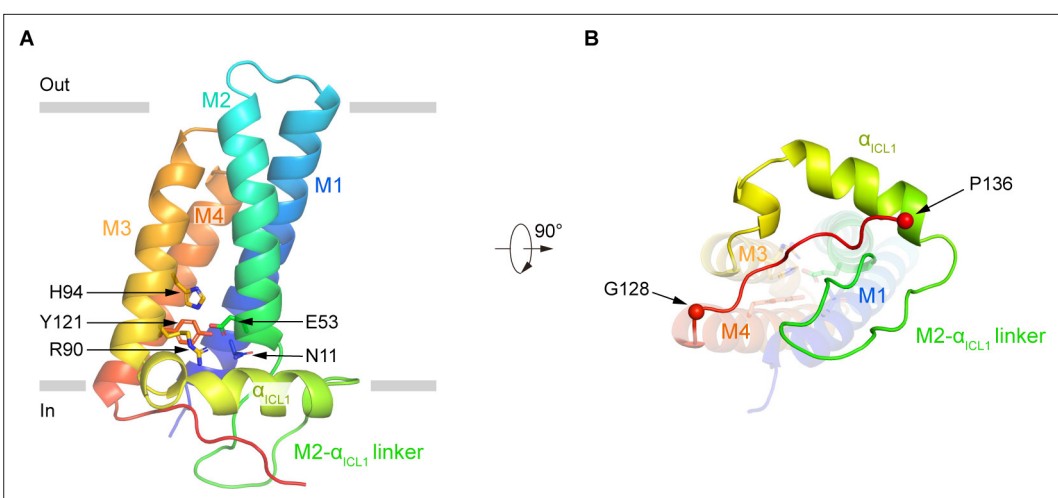

**Figure 3.** The structure of human p22. (**A**) Structure of p22 in cartoon representation, and is colored in the rainbow pattern (N-terminus is blue and C-terminus is red). The phospholipid bilayer is shown as gray layers. (**B**) The 90°-rotated bottom view of A. The 128–136 segment is colored in red. Cα positions of G128 and P136 are indicated as spheres.

## Interface between NOX2 and p22

The TMD of p22 makes extensive interactions with the TMD of NOX2, with a 9618 Å² interaction interface (*Figure 4A–C*). The assembly between NOX2 and p22 is mainly mediated by hydrophobic interactions, which involve M1 and M4 of p22 and M3, M4, and M5 of NOX2 (*Figure 4D*). Specifically, I4, W6, M8, W9, I20, V27, F33 on M1, L105, I108, L109, A112, I116 on M4 of p22 and I117, L120, F121 on M3, L167, A170, V180, L184, I187, L188 on M4, I197, Y201 on Loop D, V204 on M5 of NOX2 form two complementary shapes in the TMD of p22 and NOX2 for complex assembly (*Figure 4D*). These residues are conserved in NOX1-4 and p22 (*Figure 1—figure supplements 4–5*). Moreover, we observed lipid densities between NOX2 and p22, including a putative phosphatidylcholine molecule (*Figure 4E–F*), but their roles in the heterodimer formation remain elusive.

## Structural differences of DH domain between resting NOX2 and active DUOX1

Since the current structure represents the NOX2-p22 complex in an inactive resting state, we sought to understand how the complex might be activated. Because DUOX1 in the high-calcium state is active (*Wu et al., 2021*), we used it as the reference of the 'active DUOX1' for structural comparison. Alignment using their TMD shows that their DH domains have both rotational and translational movements relative to each other (*Figure 5A–B*). Domain motion analysis using Dyndom (*Hayward and Lee, 2002*) revealed that the rotational axis of the DH domains is tilted on the membrane and there is a 49.4° rotation and a 2.3 Å translation of the DH domain along the axis (*Figure 5A–B*). As a result, there is a concomitant 12.4 Å movement of FAD bound on DH domain (*Figure 5C*). Although the DH domain of resting NOX2 is still under and adjacent to its TMD, the orientation of FAD seems to be not optimal for efficient electron transfer to the inner haem, compared to DUOX1. Notably, several CGD mutations on the DH domain of NOX2, including S333P (*Rae et al., 1998*), G389A (*Bolscher et al., 1991*), and G389E (*Ishibashi et al., 2000*), are close to the DH domain-TMD interface (*Figure 1—figure supplement 7*), emphasizing the importance of this interface in the activity of NOX2 complex.

## Discussion

Although the common architecture of the NOX enzyme core and how they transfer electrons from intracellular NADPH to extracellular oxygen are emerging from previous structural studies (*Magnani et al., 2017*; *Sun, 2020*; *Wu et al., 2021*), the structure of the essential auxiliary subunit p22 and how p22 assembles with NOX1-4 remain elusive prior to our current study. Here, we provide the structure of the human NOX2-p22 heterodimer in the resting state at high resolution. Since the residues on

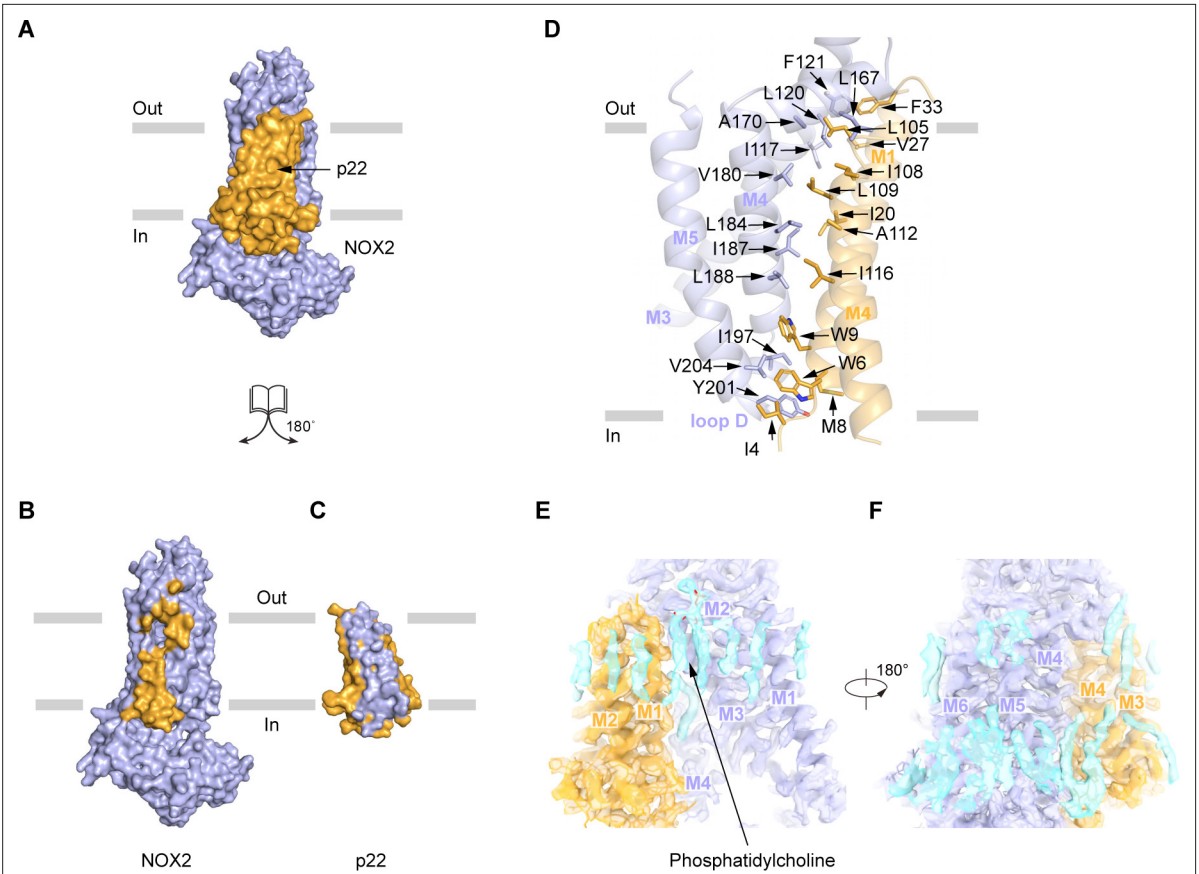

**Figure 4.** The interaction between NOX2 and p22. (**A–C**) The open-book view of the interface between NOX2 and p22 in surface representation. NOX2 and p22 are colored in light blue and orange respectively in (**A**). Residues on NOX2 interacting with p22 are colored in orange in (**B**). Residues on p22 interacting with NOX2 are colored in light blue in (**C**). The phospholipid bilayer is shown as gray layers. (**D**) The interface between NOX2 and p22 in cartoon representation. Residues participating in the interaction between NOX2 and p22 are shown as sticks. (**E**) Putative lipids that are close to NOX2-p22 interface. The map is contoured at 5.5 σ. (**F**) A 180°-rotated view of E.

NOX2 that interact with p22 are mostly conserved in NOX1-4 (*Figure 1—figure supplement 4*), we speculate that NOX1, NOX3, and NOX4 likely interact with p22 in the same manner. The interaction mode between NOX2 and p22 is drastically distinct from that of DUOX1 and DUOXA1 (*Figure 6—figure supplement 1*). DUOXA1 interacts with DUOX1 mainly through extracellular domains, including the CLD of DUOXA1 and the PHD of DUOX1. Moreover, DUOXA1 packs at the peripheral of M5 and M6 of DUOX1 (*Figure 6—figure supplement 1*), whereas p22 packs against M3 and M4 of NOX2 (*Figure 6—figure supplement 1*). Interestingly, the position of p22 is similar to the M0 of DUOX1 (*Figure 6—figure supplement 1*). The C-terminus of M0 connects the intracellular regulatory PHLD and EF domains (*Wu et al., 2021*) and the C-terminus of p22 is the binding site of cytosolic factor p47 for NOX2 activation. Therefore, both DUOX1 M0 and p22 recruit important regulatory modules of NOX, suggesting that this position might be a common binding site for the regulatory component of the NOX enzyme.

The NOX2-p22 structure presented here is in the resting state. The edge-to-edge distances between haems and the electron-relay F215 are in a suitable range for electron transfer. But the distance between inner haem and FAD (6.1 Å) is much larger than that of DUOX1 in the high-calcium state (3.9 Å). Although it is possible that protein residues or certain solvent molecules between inner haem and FAD could relay electrons between them, given the sequence and structural similarity between NOX2 and DUOX, it is highly likely that FAD directly transfers electrons to inner haem in the active NOX2 and the electron transfer from FAD to inner haem in the current structure is inefficient. Moreover, we did not observe the NADPH density in the electron density map, even when excess NADPH was present in the sample, suggesting that the NADPH affinity of NOX2 in

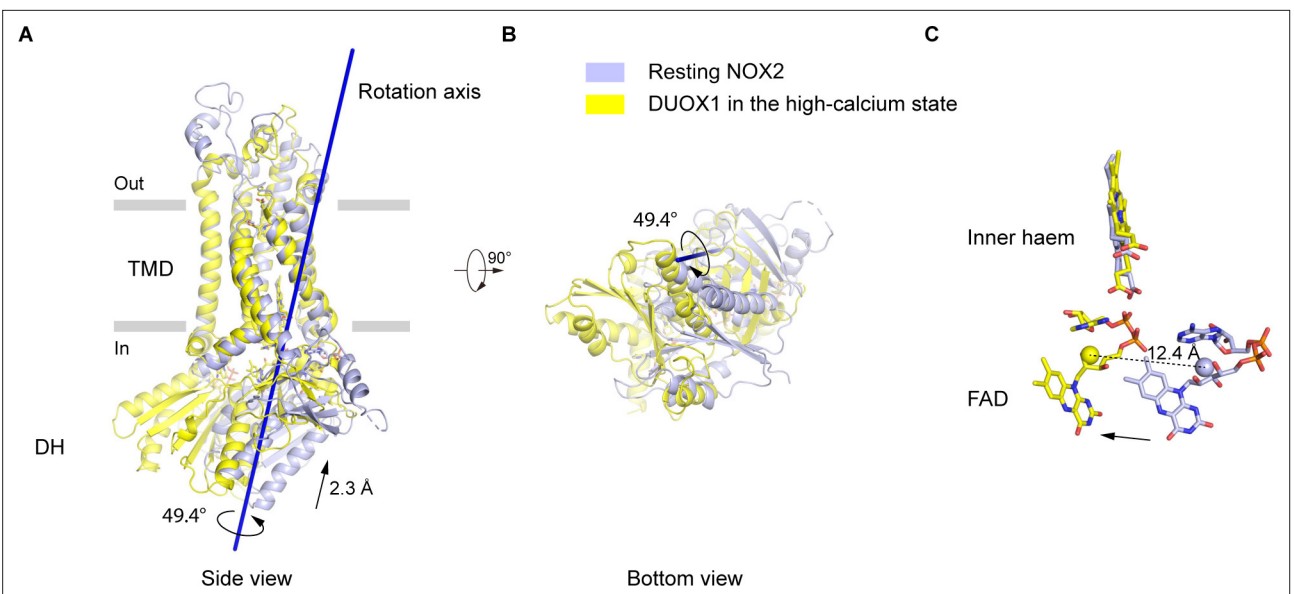

**Figure 5.** The movements of the dehydrogenase (DH) domain from the resting NOX2 to the active DUOX1. (**A**) Side view of the structural alignment of NOX2 in the resting state (light blue) and DUOX1 in the high-calcium state (yellow) in cartoon representation. Their transmembrane domains (TMD) were used in alignment. The rotation axis of DH domains is shown in dark blue. The rotational and translational movements from the DH domain of NOX2 to the DH domain of DUOX1 are indicated with arrows. (**B**) Bottom view of A. (**C**) The centers of mass of FAD molecules in the resting NOX2 and in the active DUOX1 are shown as spheres and the movement of FAD is indicated by an arrow.

the resting state is rather low. This is in contrast to the $K_m$ of NADPH in the fully activated state, which is about 37 μM (**Bromberg and Pick, 1985**), indicating that the affinity of NADPH is enhanced during NOX2 activation. Previous structural studies on DUOX1 **Sun, 2020**; **Wu et al., 2021** have shown that optimal NADPH binding involves residues from both TMD and DH domain, whereas DH domain of the resting NOX2 has an obvious displacement compared to DUOX1, indicating that NADPH could not interact with DH domain and TMD simultaneously in this structure, resulting in the low NADPH affinity in the resting state. Because DH domain of NOX2 in the resting state has obvious displacement compared to the active DUOX1, we designate the current conformation of DH domain as 'undocked', and we speculate that during the activation of NOX2, the cytosolic factors, including p47[phox], p67[phox], and Rac1, might cooperatively stabilize the DH domain of NOX2 in the 'docked' conformation, which is similar to that observed in the active DUOX1 structure, to enhance the NADPH affinity and electron transfer efficiency from FAD to inner haem to support the redox activity of NOX2 (**Figure 6**).

Notably, during the review process of this manuscript, another group published a core structure of the human NOX2-p22 and 7G5 Fab complex (abbreviated as 7G5-NOX2 complex) (**Noreng et al., 2022**). Through structural comparison, we found that the structures of the core region of the NOX2-p22 complex (NOX2 TMD and p22) are essentially the same (**Figure 6—figure supplement 2**) with RMSD of 0.364 Å, but 7D5 and 7G5 Fab bind to an overlapped but not identical structural epitope of NOX2. 7G5 only binds to loop E while 7D5 binds to both loop C and loop E. Because these structures are determined using different antibody Fab fragments, their high structural similarity in NOX2-p22 core demonstrates that these Fab fragments do not affect the structure of the NOX2-p22 core. This is in agreement with the fact that neither 7G5 Fab (**Noreng et al., 2022**) nor 7D5 Fab (**Figure 1B**) affects the enzymatic activity of NOX2 in the in vitro assay. The fundamental difference between these two structures is in the DH domain of NOX. The DH domain of the 7G5-NOX2 complex determined in the detergent micelle is largely disordered while the DH domain of our NOX2 structure in nanodisc could be readily observed and refined to 3.1 Å resolution. This sharp contrast is likely because the lipidic environment provided by nanodisc is essential for the structural stability of DH domain and this correlates with the previous data showing that highly purified NOX2 in detergent but without lipids has no activity (**Shpungin et al., 1989**; **Harper et al., 1984**).

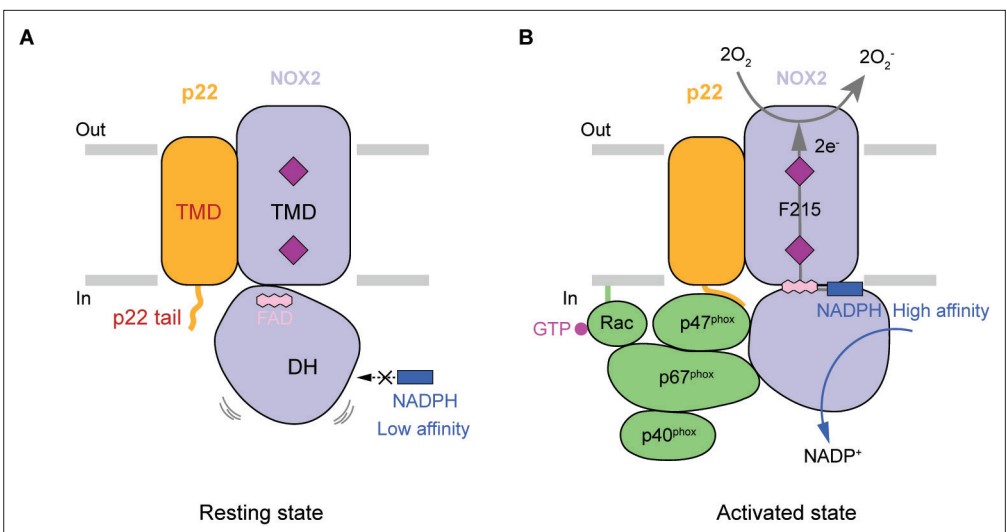

**Figure 6.** Hypothetic model for NOX2 activation. (**A**) Resting state of the phagocyte NADPH oxidase. NOX2, p22, and cytosolic factors are shown as cartoon and colored the same as *Figure 1A*. (**B**) Activated state of the phagocyte NADPH oxidase. The electron transfer pathway is indicted with gray arrows.

The online version of this article includes the following figure supplement(s) for figure 6:

**Figure supplement 1.** Distinct interaction mode between NOX2-p22 complex and DUOX1-DUOXA1 complex.

**Figure supplement 2.** Structural comparison between NOX2-7D5 complex and NOX2-7G5 complex.

Taken together, our structure of functional human NOX2-p22 complex in the resting state reveals the assembly mechanism of this important heterodimer and provides an essential reference to interrogate the activation mechanism of NOX2 by cytosolic factors.

# Materials and methods

## Key resources table

| Reagent type (species) or resource | Designation | Source or reference | Identifiers | Additional information |
|---|---|---|---|---|
| Gene (*Homo sapiens*) | CYBA | GenBank | AB590173.1 | |
| Gene (*Homo sapiens*) | CYBB | GenBank | NM_000397.4 | |
| Gene (*Homo sapiens*) | p47phox | GenBank | NM_000265.7 | |
| Gene (*Homo sapiens*) | p67phox | GenBank | LT740922.1 | |
| Gene (*Homo sapiens*) | rac1 | GenBank | NM_006908.5 | |
| Cell line (*Spodoptera frugiperda*) | Sf9 | Thermo Fisher Scientific | Cat. # 12659017 | RRID:CVCL_0549 |
| Cell line (*Homo sapiens*) | FreeStyle 293F | Thermo Fisher Scientific | Cat. # R79007 | RRID:CVCL_D603 |
| Transfected construct (human) | NOX2-p22 complex | This paper | | Transfected construct (human) |
| Antibody | Anti-NOX2 7D5 (Mouse monoclonal) | MBL Life Science | Cat. # D162-3 | |
| Recombinant DNA reagent | p22-GFP (plasmid) | This paper | | GFP version of p22phox |
| Recombinant DNA reagent | GFP-TP1170 (plasmid) | This paper; *Pleiner et al., 2018* | PMID:29263082 | GFP version of TP1170 |
| Peptide, recombinant protein | SOD | Sigma-Aldrich | Cat. # S2515 | |

*Continued on next page*

*Continued*

| Reagent type (species) or resource | Designation | Source or reference | Identifiers | Additional information |
|---|---|---|---|---|
| Commercial assay or kit | In-Fusion HD Cloning | Takara Bio | Cat. # 639650 | |
| Chemical compound, drug | Amplex Red | GeneCopoeia | Cat. # C291 | |
| Chemical compound, drug | NADPH | Cayman Chemical | Item No. 9000743 | |
| Software, algorithm | SWISS-MODEL | SWISS-MODEL | RRID:SCR_013032 | |
| Software, algorithm | Chimera | Chimera | RRID:SCR_004097 | |
| Software, algorithm | Coot | Coot | RRID:SCR_014222 | |
| Software, algorithm | Phenix | Phenix | RRID:SCR_014224 | |

## Cell culture

Sf9 insect cells (Thermo Fisher Scientific) were cultured in Sf-900 III serum-free medium (Gibco) or SIM SF serum-free medium (Sino Biological) at 27°C. HEK293F suspension cells (Thermo Fisher Scientific) were cultured in FreeStyle 293 medium (Thermo Fisher Scientific) supplemented with 1% fetal bovine serum at 37°C with 6% $CO_2$ and 70% humidity. The cell lines were routinely checked to be negative for mycoplasma contamination but have not been authenticated.

## Enzymatic assay

The superoxide anion-generating activities of the crude cell lysates containing membranes of NOX2-p22 complex were measured using the Amplex Red assay (*Zhou et al., 1997*). The concentrations of $H_2O_2$ solution were determined by measuring UV-Vis absorbance at 240 nm with spectrophotometer (Pultton) and calculated using molar extinction coefficient of 43.6 $M^{-1}$ $cm^{-1}$. The concentration of $H_2O_2$ solution was further validated by reacting with Amplex Red to generate resorufin which has $\varepsilon_{571}$=69,000 $M^{-1}$ $cm^{-1}$ *Zhou et al., 1997*. Then, the $H_2O_2$ solution with known concentration was used to calibrate the standard resorufin fluorescence curve (excitation, 530 nm; emission, 590 nm) measured using a Microplate Reader (Tecan Infinite M Plex) at 30°C. The reactions were performed at 30°C in 0.15 ml of HBS (20 mM HEPES, pH 7.5, 150 mM NaCl) with 1 mM $MgCl_2$, 1 mM EGTA, 10 µM FAD, 50 µM NADPH, 25 µM Amplex Red, 0.067 mg/ml horseradish peroxidase and 0.0576 mg/ml SOD. Thirty-three nM Trimera was added to activate NOX2 activity and 10 µg/ml DPI was added to inhibit NOX2 activity. Progress of the reactions was monitored continuously by following the increase of the resorufin fluorescence, and the initial reaction rates were obtained by fitting the curve with linear equation. The data was processed with Microsoft Excel 2013 and GraphPad Prism 6.

The superoxide anion-generating activity of the NOX2-p22-7D5-TP1170 complex in nanodisc was determined essentially the same as described above. The reaction system contains 0.36 nM NOX2-p22-7D5-TP1170 complex in nanodisc and no amphiphile was added into the system.

## Protein expression, purification, and nanodisc reconstitution

Gene encoding human p47$^{phox}$ (1-286), human p67$^{phox}$ (1-226), and full-length human Rac1Q61L were fused together using the In-Fusion HD Cloning Kit (Takara Bio) and linked into a pET-based vector containing N-terminal 6×His tag. The sequence of the linker between p47$^{phox}$ and p67$^{phox}$ was AAAST-GGGSS, and the sequence of the linker between p67$^{phox}$ and Rac1Q61L was GGGGGG. His6-Trimera protein was expressed in BL21 (DE3) and purified using Talon resin and HiTrap Q HP anion exchange chromatography column (GE Healthcare).

Mouse anti-human NOX2 monoclonal antibody 7D5 (*Yamauchi et al., 2001*) was purchased from MBL Life Science. 7D5 Fab was released by papain digestion. Gene encoding TP1170 (*Pleiner et al., 2018*) was synthesized and cloned into a pET-based vector containing N-terminal GFP tag and strep tag. GFP-strep-TP1170 protein was purified from Rosetta-gami2 (DE3).

Human p22 was fused with C-terminal GFP tag and strep tag and cloned into the pBMCL1 vector to generate pBMCL1-p22 (*Guo et al., 2022*). Human NOX2 was cloned into a modified BacMam vector (*Li et al., 2017*) and linked into pBMCL1-p22 using LINK method (*Scheich et al., 2007*). The corresponding BacMam virus was made using sf9 cells (*Wu et al., 2021*). For protein expression, HEK293F cells cultured in Free Style 293 medium at a density of $2.5 \times 10^6$ cells per ml were infected

with 10% volume of P2 virus. Sodium butyrate (10 mM) and 5-aminolevulinic acid hydrochloride (200 µM) was added to the culture 12 hr after transfection to promote haem incorporation, and the cells were transferred to a 30°C incubator for another 36 hr before harvesting. Cells were collected by centrifugation at 3999 × g (JLA-8.1000, Beckman Coulter) for 10 min at 4°C, and washed with TBS buffer (20 mM Tris pH 8.0 at 4°C, 150 mM NaCl) containing 2 µg/ml aprotinin, 2 µg/ml pepstatin, 2 µg/ml leupeptin. The cells were then flash-frozen and stored at –80°C. For purification, cell pellets were resuspended in Buffer A (20 mM HEPES pH 7.5, 150 mM NaCl, 2 µg/ml aprotinin, 2 µg/ml pepstatin, 2 µg/ml leupeptin, 20% (v/v) glycerol, 1 mM DTT) containing 1 mM phenylmethanesulfonyl fluoride, 2 mM EGTA, 10 mM MgCl$_2$, 10 µg/ml DNAse, 19.6 mM dodecylmaltoside (DDM), and 1.65 mM cholesteryl hemisuccinate. The mixture was stirred for 1 hr at 4°C to solubilize membrane proteins. The insoluble debris was removed by centrifugation at 193,400 × g (Ti50.2, Beckman Coulter) for 40 min. Subsequently, the supernatant was loaded onto 4 ml Streptactin Beads 4FF (Smart Life-sciences) column and washed with 50 ml wash buffer 1 (Buffer A supplemented 0.5 mM DDM, 10 mM MgCl$_2$ and 2 mM adenosine triphosphate) to remove contamination of heat shock proteins. Then, the column was extensively washed with 50 ml wash buffer 2 (Buffer A supplemented with 0.5 mM DDM). The target protein was eluted with Buffer A supplemented 0.5 mM DDM, and 8.5 mM D-desthio-biotin (IBA). The eluate was loaded onto HiTrap Q HP (GE Healthcare) and the NOX2-p22 complex was separated from aggregates with a linear gradient from 0 mM NaCl to 1000 mM NaCl in buffer containing 20 mM HEPES pH 7.5 at 4°C, 0.5 mM DDM. The fractions containing the NOX2-p22 complex were collected for nanodisc reconstitution. NOX2-p22 complex was mixed with MSP2X and L-α-phosphatidylcholine from soybean (PC, Type II-S, 14–23% choline basis, Sigma P5638, solu-bilized in 68.49 mM octyl glucoside) at a molar ratio of protein: MSP2X: PC = 1:4:800. The mixture was allowed to equilibrate for 1 hr at 4°C, and then Bio-beads SM2 (Bio-Rad) was added to initiate the reconstitution with constant rotation at 4°C. Bio-beads SM2 were added to the mixture three times within 15 hr to gradually remove detergents from the system. Afterward, the proteins that were not reconstituted in lipid nanodisc was removed by centrifugation at 86,600 × g for 30 min in TLA 100.3 rotor (Beckman Coulter). The supernatant containing NOX2-p22 complex reconstituted in lipid nanodisc was loaded onto the 1 ml Streptactin Beads 4FF column to remove empty nanodisc. The elution from Streptactin Beads 4FF column was concentrated and subjected onto a Superose 6 increase 10/300 GL column (GE Healthcare) in buffer that contained 20 mM HEPES pH 7.5, 150 mM NaCl. The peak fractions corresponding to the NOX2-p22 complex in lipid nanodisc were collected. The GFP tag at the C-terminus of p22 was cleaved off by Prescission Protease digestion. Then 7D5 Fab and GFP-strep-TP1170 protein was added into NOX2-p22 nanodisc sample and the mixture was subjected to Streptactin Beads 4FF to remove excess p22 and free Fc fragment. NOX2-p22-7D5-TP1170 quaternary complex was eluted with D-desthiobiotin, concentrated and subjected to Superose 6 increase 10/300 GL column. Peak fractions containing the complex were pooled for cryo-EM sample preparation. Concentration of NOX2 was estimated with absorbance at 415 nm (ε$_{415}$=0.262 µM$^{-1}$ cm$^{-1}$) (*Jesaitis et al., 2019*).

## Cryo-EM sample preparation and data collection

To obtain a homogenous orientation distribution of particles, the surface of Quantifoil Au 300 mesh R 0.6/1.0 grids were coated with graphene oxide (GO grids) (*Patel et al., 2021*) or graphene (G grids) (*Zhang et al., 2017*). Aliquots of 2.5 µl of NOX2-p22-7D5-TP1170 complex in nanodisc at a concentration of approximately 0.12 mg/ml in the presence of 0.5 mM FAD, 1 mM NADPH, and 0.5 mM fluorinated octyl-maltoside (FOM, Anatrace) were applied to the grids. After 60 s incubation on the grids at 4°C under 100% humidity, the grids were then blotted for 4 s using a blot force of 4, and then plunge-frozen into liquid ethane using a Vitrobot Mark IV (Thermo Fisher Scientific). The grids were transferred to a Titan Krios electron microscope (Thermo Fisher) operating at 300 kV and a K2 Summit direct detector (Gatan) mounted post a quantum energy filter (slit width 20 eV). SerialEM-3.6.11 was used for automated data collection. Movies from dataset of NOX2-p22 were recorded in super-resolution mode and a defocus range of –1.5 to –2.0 µm with a nominal magni-fication of ×165,000, resulting in a calibrated pixel size of 0.821 Å. Each stack of 32 frames was exposed for 8 s, with an exposing time of 0.25 s per frame at a dose rate of 4.7 electrons per Å$^{-2}$ per second.

## Image processing

A total of 1890 and 2059 micrographs of the NOX2-p22 complex on graphene oxide-coated grid and graphene-coated grid were collected, separately. The beam-induced drift was corrected using MotionCor2 (*Zheng et al., 2017*) and binned to a pixel size of 1.642 Å. Dose-weighted micrographs were used for CTF estimation by Gctf (*Zhang, 2016*). A total of 206,504 and 355,928 particles were auto-picked using Gautomatch-0.56 (developed by K Zhang) or Topaz-0.2.3 (*Bepler et al., 2019*) and extracted with the box size of 280 pixels. Two rounds of 2D classification in Relion (*Zivanov et al., 2018*) were performed to remove ice contaminants and aggregates, yielding 14,287 and 27,177 particles and used as seeds for seed-facilitated classification (*Wang et al., 2021*) with cryoSPARC (*Punjani et al., 2017*). Then, two datasets were combined and duplicated particles were removed, and 208,906 particles were retained and re-extracted with box size of 320 pixels and rescaled to 280 pixels to generate a pixel size of 1.0556 using cryoSPARC. The re-extracted and re-scaled particles were subjected to multi-reference 3D classification using resolution gradient (*Wang et al., 2021*). The resulting particles of selected classes were subjected to non-uniform refinement and local refinement to generate a consensus map at 2.8 Å resolution (*Punjani et al., 2020*). To further improve the local map quality of extracellular TP1170-constant region of Fab fragment (mask1), intracellular DH domain (mask2), and the NOX2 core region (mask3), these regions were subjected to signal subtraction and local refinement in cryoSPARC to generate three local maps with average resolution of 3.1, 3.1, and 2.8 Å. These three focus refined maps were aligned to the consensus map and merged to generate a composite map which was further cropped to the box size of 190 for interpretation and model building. All resolutions were estimated using the gold-standard Fourier shell correlation 0.143 criterion. Local resolution was calculated using cryoSAPRC.

## Model building

Predicted models of NOX2 and p22 were downloaded from Alpha Fold 2 database (*Jumper et al., 2021*). Fab was modeled as poly-alanine. Model of TP1170 was modeled using SWISS-MODEL (*Biasini et al., 2014*). Individual models were docked into the map using Chimera (*Pettersen et al., 2004*). The models were iteratively manually rebuilt using Coot (*Emsley et al., 2010*) according to the map and refined against the map using Phenix (*Adams et al., 2010*; *Afonine et al., 2018*).

## Quantification and statistical analysis

The number of independent reactions (N) and the relevant statistical parameters for each experiment (such as mean or standard deviation) are described in the figure legends. No statistical methods were used to pre-determine sample sizes.

## Acknowledgements

We greatly thank Miao Wei for providing the GFP-strep-TP1170 protein. We thank Xiaoyu Liu and Yiting Shi for illustration. Cryo-EM data collection was supported by Electron microscopy laboratory and Cryo-EM platform of Peking University with the assistance of Xuemei Li, Zhenxi Guo, Changdong Qin, Xia Pei, Xiaojuan Hui, and Guopeng Wang. Part of the structural computation was also performed on the Computing Platform of the Center for Life Science and High-performance Computing Platform of Peking University. We thank the National Center for Protein Sciences at Peking University in Beijing, China, for assistance with negative stain EM. The work is supported by grants from the Ministry of Science and Technology of China (National Key R&D Program of China, 2022YFA1303000 to LC), National Natural Science Foundation of China (91957201, 31870833, and 31821091 to LC; 52021006 to HP), Beijing National Laboratory for Molecular Sciences (BNLMS-CXTD-202001 to HP), the National Key Research and Development Program of China, and Center for Life Sciences (CLS to LC).

## Additional information

### Funding

| Funder | Grant reference number | Author |
|--------|------------------------|--------|
| National Key Research and Development Program of China | 2022YFA1303000 | Lei Chen |
| National Natural Science Foundation of China | 91957201 | Lei Chen |
| Beijing National Laboratory for Molecular Sciences | BNLMS-CXTD-202001 | Hailin Peng |
| National Natural Science Foundation of China | 52021006 | Hailin Peng |
| National Natural Science Foundation of China | 31821091 | Lei Chen |
| National Natural Science Foundation of China | 31870833 | Lei Chen |
| Center for Life Sciences | CLS | Lei Chen |

The funders had no role in study design, data collection and interpretation, or the decision to submit the work for publication.

### Author contributions

Rui Liu, Kangcheng Song, Conceptualization, Data curation, Formal analysis, Validation, Investigation, Visualization, Writing – review and editing; Jing-Xiang Wu, Conceptualization, Formal analysis, Investigation; Xiao-Peng Geng, Liming Zheng, Xiaoyin Gao, Investigation; Hailin Peng, Conceptualization, Resources, Supervision, Funding acquisition, Methodology, Project administration; Lei Chen, Conceptualization, Resources, Data curation, Formal analysis, Supervision, Funding acquisition, Validation, Visualization, Methodology, Writing – original draft, Project administration, Writing – review and editing

### Author ORCIDs

Rui Liu ⬤ http://orcid.org/0000-0002-3758-6493
Kangcheng Song ⬤ http://orcid.org/0000-0001-7932-2202
Jing-Xiang Wu ⬤ http://orcid.org/0000-0001-9851-0065
Lei Chen ⬤ http://orcid.org/0000-0002-7619-8311

### Decision letter and Author response

Decision letter https://doi.org/10.7554/eLife.83743.sa1
Author response https://doi.org/10.7554/eLife.83743.sa2

## Additional files

### Supplementary files
• MDAR checklist

### Data availability
Cryo-EM maps and atomic coordinate of the NOX2-p22-7D5-TP1170 complex have been deposited in the EMDB and PDB under the ID codes EMDB: EMD-34389 and PDB: 8GZ3.

The following datasets were generated:

| Author(s) | Year | Dataset title | Dataset URL | Database and Identifier |
|---|---|---|---|---|
| Liu Rui, Chen Lei | 2022 | Structure of human NOX2 resting state | https://www.rcsb.org/structure/8GZ3 | RCSB Protein Data Bank, 8GZ3 |
| Liu Rui, Chen Lei | 2022 | Structure of human NOX2 in the resting state | https://www.ebi.ac.uk/emdb/EMD-34389 | EMDB, EMD-34389 |

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
