## [Editor Report]

NADPH oxidases are a family of membrane enzymes that produce reactive oxygen species (ROS). NOX2 is the most well-studied member of the NADPH oxidase family, and the proper function of NOX2 is critical for innate immunity against pathogens in mammals. The study by Dr. Chen and colleagues used antibodies to facilitate the structural determination of the high-resolution structure of the NOX2-p22 complex, which is otherwise challenging for single-particle analysis due to its flexibility and relatively small molecular weight. This structural study provides valuable information for a mechanistic understanding of NOX2 activation at the molecular level.

---

## [Decision Letter]

**Decision letter after peer review:**

Thank you for submitting your article "Structure of human phagocyte NADPH oxidase in the resting state" for consideration by *eLife*. Your article has been reviewed by 3 peer reviewers, and the evaluation has been overseen by a Reviewing Editor and Volker Dötsch as the Senior Editor. The following individual involved in review of your submission has agreed to reveal their identity: Edgar Pick (Reviewer #3).

Essential revisions:

1. A recent paper appeared (Noreng S et al., PMID: 36241643), describing the structure of the NOX2-p22 complex in the presence of an inhibitory antibody 7G5. In this paper, the authors did not compare the function of NOX2 in the presence and absence of the antibody 7D5. It is unclear how NOX2 interacts with 7D5 in atomic details, and whether there is any difference in the NOX2-antibody interaction between the 7D5 antibody and the 7G5 antibody used in the study by Noreng S et al. Please discuss in depth the similarities and differences between the structures of Noreng et al., and the present paper, commenting on the interaction between NOX2 and p22, between the DH domain and transmembrane domain of NOX2, and between the antibodies and NOX2 extracellular loop. Please include a comparison with structures of other NADPH oxidases.

2. Lipids are critical for NOX2 function (PMID: 8392946 and PMID: 2542302). Notably, In the study by Noreng S et al., lipids are important in mediating NOX2 and p22 interaction. Please comment if any lipid molecule(s) are observed in the NOX2 and p22 complex and validate if unique interactions exist.

3. A major finding of this paper is the conformational flexibility of DH in NOX2 in the absence of cytosolic partners. Please provide a detailed structural description of the difference and its functional significance (eg. mapping disease mutation at the interface, if any).

4. The authors imply that cytosolic domains could stabilize the DH domain of NOX2 and thus contribute to its functional activation. Should one expect that the NOX2 DH domain would be more stable upon Trimera binding? Do the authors have any data to support this statement?

5. The manuscript could more deeply analyze the catalytic site around the outer heme. What is the size of the site? Would an oxygen fit inside? The text mentions a tunnel leading to the site, but figures 2 C and D are unclear.

6. Based on Figure 2, the flavin is at about 7-10 Angstroms distance from the inner heme. That distance is fully compatible with an electron transfer process. Please specifically describe this site. Is there solvent between the flavin and the heme? Could one think of a protein- and FAD-based conduct allow electrons to move from the flavin to the heme?

7. Superoxide is probably not the "direct" killer of pathogens but superoxide-derived reactive oxygen species (ROS), such as H2O2, hydroxyl radicals and singlet oxygen are responsible. Please comment.

8. There are Ramachandran outliers, clashes, and misfitting sidechains reported in the PDB validation report. Please improve the atomic model.

---

## [Author Response]

Essential Revisions (for the authors):1. A recent paper appeared (Noreng S et al., PMID: 36241643), describing the structure of the NOX2-p22 complex in the presence of an inhibitory antibody 7G5. In this paper, the authors did not compare the function of NOX2 in the presence and absence of the antibody 7D5. It is unclear how NOX2 interacts with 7D5 in atomic details, and whether there is any difference in the NOX2-antibody interaction between the 7D5 antibody and the 7G5 antibody used in the study by Noreng S et al. Please discuss in depth the similarities and differences between the structures of Noreng et al., and the present paper, commenting on the interaction between NOX2 and p22, between the DH domain and transmembrane domain of NOX2, and between the antibodies and NOX2 extracellular loop. Please include a comparison with structures of other NADPH oxidases.

We have provided new experimental results showing that 7D5 Fab does not affect the enzymatic activity of NOX2 in vitro in Figure 1B of the revised manuscript. We have included the structural comparison of our structure and previously published csNOX5 and DUOX1 in Figure 1—figure supplement 5. The structure comparison between NOX2-7D5 and NOX2-7G5 complex is shown in Figure 6—figure supplement 2, showing that the NOX2-p22 regions of these two structures are almost identical. The structural epitopes of 7D5 and 7G5 on NOX2 overlap but are not identical. Notably, the DH domain was not observed in the NOX2-7G5 structure by Noreng et al., but was observed in our structure. We have included these discussions in the revised manuscript.

2. Lipids are critical for NOX2 function (PMID: 8392946 and PMID: 2542302). Notably, In the study by Noreng S et al., lipids are important in mediating NOX2 and p22 interaction. Please comment if any lipid molecule(s) are observed in the NOX2 and p22 complex and validate if unique interactions exist.

We observed many lipid-like densities in our maps. We have shown them in the revised Figure 1E-F and Figure 4E-F. We think it is likely that these lipids might play important functional roles in mediating NOX2-p22 interactions or the function of NOX2, but their exact functions require further functional evidences which are lacking at the moment. Therefore, we did not expend our discussion on them to avoid over-interpret our results.

3. A major finding of this paper is the conformational flexibility of DH in NOX2 in the absence of cytosolic partners. Please provide a detailed structural description of the difference and its functional significance (eg. mapping disease mutation at the interface, if any).

We apologize with the confusion about the DH domain flexibility. Although the DH domain has an obvious degree of flexibility as observed in the consensus map, we would like to emphasize that the DH domain is “undocked” to the TMD and is not optimal for electron transfer and NADPH binding. Please see revised Figure 6 for the new model and related revised discussion. We have also included the description of CGD mutations that are close to the DH-TMD interface at line 198.

4. The authors imply that cytosolic domains could stabilize the DH domain of NOX2 and thus contribute to its functional activation. Should one expect that the NOX2 DH domain would be more stable upon Trimera binding? Do the authors have any data to support this statement?

We speculate that the binding of Trimera would reorient the DH domain and activate NOX2, as shown in the revised Figure 6. This is a speculation based on the structure of NOX2 in the resting state and the structure of DUOX1 in the high-calcium state. We do not have experimental evidence yet.

5. The manuscript could more deeply analyze the catalytic site around the outer heme. What is the size of the site? Would an oxygen fit inside? The text mentions a tunnel leading to the site, but figures 2 C and D are unclear.

We have provided revised Figure 2C-F to better depict the oxygen-reducing center and the tunnel connecting it. We found a putative water density at the oxygen-reducing center, which is large enough for oxygen, too.

6. Based on Figure 2, the flavin is at about 7-10 Angstroms distance from the inner heme. That distance is fully compatible with an electron transfer process. Please specifically describe this site. Is there solvent between the flavin and the heme? Could one think of a protein- and FAD-based conduct allow electrons to move from the flavin to the heme?

We have provided a revised Figure 2H for better visualization of the FAD binding pocket. In the structure of active DUOX1, FAD transfer electrons directly to the inner haem. Given the sequence and structural similarity between NOX2 and DUOX1, it is highly likely that FAD of activated NOX2 also transfers electrons to the inner haem directly. We observed that F202 on M5 of NOX2 is between FAD and inner haem, but we do not have any evidence showing that F202 could mediate electron transfer between FAD and inner haem and we prefer to stay cautious about such speculation.

7. Superoxide is probably not the "direct" killer of pathogens but superoxide-derived reactive oxygen species (ROS), such as H2O2, hydroxyl radicals and singlet oxygen are responsible. Please comment.

We have revised the manuscript to avoid the expression that superoxide anions kills pathogens directly.

8. There are Ramachandran outliers, clashes, and misfitting sidechains reported in the PDB validation report. Please improve the atomic model.

We have improved our structural model based on our new high-resolution map. Please find the validation report uploaded.